# A general method for determining secondary active transporter substrate stoichiometry

Gabriel A Fitzgerald, Christopher Mulligan, Joseph A Mindell*

Membrane Transport Biophysics Section, Porter Neuroscience Research Center, National Institute of Neurological Disorders and Stroke, National Institutes of Health, Bethesda, United States

**Abstract** The number of ions required to drive substrate transport through a secondary active transporter determines the protein's ability to create a substrate gradient, a feature essential to its physiological function, and places fundamental constraints on the transporter's mechanism. Stoichiometry is known for a wide array of mammalian transporters, but, due to a lack of readily available tools, not for most of the prokaryotic transporters for which high-resolution structures are available. Here, we describe a general method for using radiolabeled substrate flux assays to determine coupling stoichiometries of electrogenic secondary active transporters reconstituted in proteoliposomes by measuring transporter equilibrium potentials. We demonstrate the utility of this method by determining the coupling stoichiometry of VcINDY, a bacterial $Na^+$-coupled succinate transporter, and further validate it by confirming the coupling stoichiometry of vSGLT, a bacterial sugar transporter. This robust thermodynamic method should be especially useful in probing the mechanisms of transporters with available structures.

*For correspondence: mindellj@ninds.nih.gov

**Competing interests:** The authors declare that no competing interests exist.

## Introduction

Secondary active transporters serve a wide range of physiological roles, including nutrient uptake, signal transduction, homeostatic regulation, and toxin efflux. These integral membrane proteins use established ion gradients to drive their substrates uphill, against their own gradients. Coupling stoichiometry, the stoichiometric ratio of coupling ion to substrate molecules transported per transport cycle, dictates the extent to which a substrate can be moved against its concentration gradient. By coupling the transport of one substrate molecule to several ions, a transporter can generate a large substrate gradient. For example, the glutamate transporter EAAT3 couples the cotransport of 3 $Na^+$, 1 $H^+$, and counter transport of 1 $K^+$ to glutamate uptake; in physiological salt solutions, these coupled ions can generate a $5 \times 10^6$ fold glutamate gradient (*Zerangue and Kavanaugh, 1996*). This ability to robustly clear away extracellular glutamate is critical in the mammalian CNS, where synaptically released glutamate must be removed to prepare a synapse for subsequent signaling events.

Comprehensive insight into transporter mechanisms requires accurate determination of transport stoichiometry. In a symporter, for example, transport is initiated by the binding of a full complement of coupling ions and substrates; successful mechanistic analysis, by experiment or by computation requires a clear determination of these key parameters. Many X-ray structures of transporters do not reveal all the ions and substrates required to initiate transport (*Mancusso et al., 2012*; *Reyes et al., 2009*; *Yernool et al., 2004*). Yet, the computational analyses that are becoming widespread and important tools require this information to accurately simulate the transport process. Accurately determining the coupling stoichiometry is therefore crucial to both mechanistic and computational

studies of secondary transporters, as well as providing insight into the transporters' physiological role.

Current methods for stoichiometry determination for the bacterial transporters most accessible to structure determination are inadequate. Coupling stoichiometry is often estimated by the measure of Hill coefficients, based on fitting a simple kinetic model to concentration-dependent measurements of transport rate. However, the information yielded by these methods strongly depends on the choice of kinetic model and can be misleading (*Lolkema and Slotboom, 2015*). Alternatively, stoichiometry can be assessed by analyzing the parallel uptake of radiolabeled substrate and radioactive coupling ion (*Groeneveld and Slotboom, 2010*). However, many candidate coupling ions are unavailable in radioactive form, or are difficult to use, particularly $^{22}Na^+$, which is highly radioactive and binds to many of the surfaces used in transport assays (*Groeneveld and Slotboom, 2010*).

An ideal method to measure coupling stoichiometry would be model independent and would use easily accessible materials. The measurement of reversal potentials fulfills these requirements, providing a thermodynamic route to transporter coupling ratios. In contrast to the Hill coefficient, this approach yields stoichiometry without requiring any prior knowledge of the transport mechanism. For an electrogenic secondary transporter, equilibrium occurs when the downhill diffusional 'force' due to substrate and ion gradients is exactly balanced by the voltage generated in each cycle by the separation of charge induced by substrate movement (*Figure 1*). This voltage is predicted by a simple equation that involves the magnitudes of ion and substrate gradients, and the charges and stoichiometries of the ions involved (see Materials and methods for derivation). If the voltage at the membrane deviates from this equilibrium value, substrate and ion will move in response (*Figure 1a, c*). Thus, if we apply a series of voltages to the system and observe substrate flux, in either direction

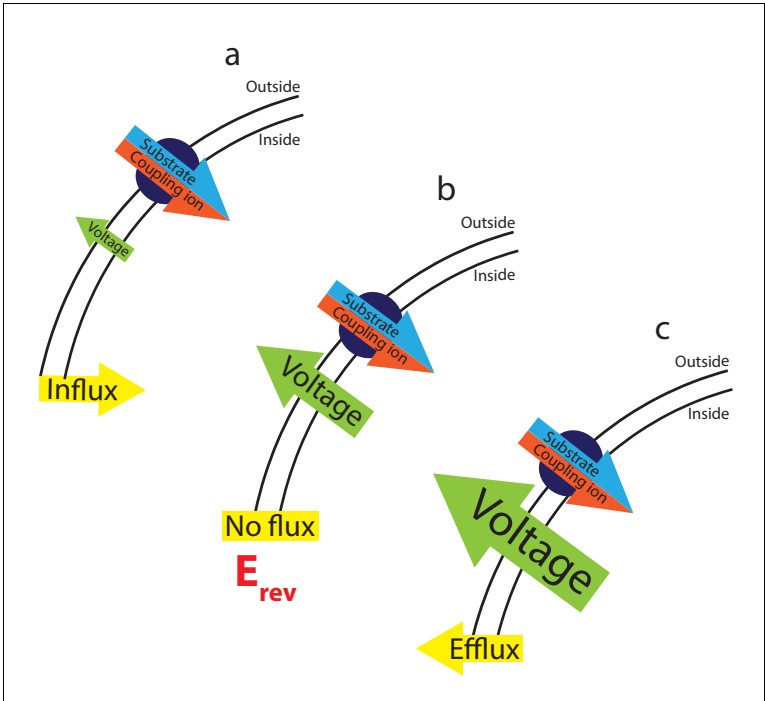

**Figure 1.** The interplay between substrate gradients, the membrane potential and flux. Electrogenic transporters cause charge build up across the membrane, which inhibits further transport. The combined gradients of coupling ion and substrate (illustrated here for a symporter with red/blue arrow), and the applied membrane potential (voltage, green arrow) therefore influence the direction of substrate flux across the membrane. Depending on the magnitude of these opposing forces, three outcomes can occur: (a) at applied voltages that are insufficient to oppose the diffusional force of the substrates, influx of substrate occurs. (b) At the equilibrium, or reversal, potential, $E_{rev}$, the applied voltage *exactly* opposes the diffusional force of the substrates resulting in no net flux. (c) At higher applied voltages, the voltage overcomes the electrochemical gradient imposed by the coupling ion and substrate gradients, reversing the direction of flux and efflux occurs.

across the membrane, the system must not be at equilibrium. At a voltage, on the other hand, where no net flux occurs, the system must be at equilibrium (*Figure 1b*). This voltage is commonly referred to as the 'reversal potential' as the ion and substrate fluxes change direction when the voltage traverses this value.

Reversal potentials are commonly measured for ion channels using electrophysiological methods in a variety of systems, including *Xenopus laevis* oocytes and patch clamped mammalian cells (*Chen et al., 1995*; *Leisle et al., 2011*; *Zerangue and Kavanaugh, 1996*). Such methods have been used for several eukaryotic transporter proteins which can be expressed in these systems and assayed using electrophysiology (*Levy et al., 1998*; *Zerangue and Kavanaugh, 1996*). However, electrophysiological measurements are not easily applied to prokaryotic transporters since these proteins generally do not express well in the eukaryotic cells where electrical recording is straightforward. For $H^+$-coupled transporters, fluorescent pH-sensitive probes have been used successfully to track $H^+$ flux and thereby determine transporter stoichiometry (*Graves et al., 2008*; *Parker et al., 2014*). However, this method is not widely applicable as fluorescent dyes sensing other ions, like $Na^+$, have not yet been successfully applied (J. Mindell, unpublished observations).

Where previous reversal potential determination methods have measured the reversal of either electrical current or coupling ion flux, another tactic would be to measure the reversal of substrate flux itself. Here, we report a novel approach to determine coupling stoichiometry of purified reconstituted transporters by determining the reversal potential of radiolabeled substrate flux. We used this method to determine the $Na^+$:substrate coupling ratio for VcINDY, a member of the divalent anion:$Na^+$ symporter (DASS) family from *Vibrio cholerae* whose structure has been determined, but for which an accurate stoichiometry is not yet known (*Mancusso et al., 2012*). We validated our method using a transporter with known stoichiometry, vSGLT, a $Na^+$/galactose symporter which transports 1 $Na^+$ per sugar (*Turk et al., 2000*). Our method is straightforward to perform, robust, and is potentially applicable to any electrogenic secondary transporter.

## Results

In a secondary active transporter, the transport process involves the movement of charge (a coupled ion, a charged substrate, or both) across the membrane, driven by changes in electrochemical potential. If the net charge moved through the entire transport cycle is nonzero, then each cycle will separate charge and add to the total membrane voltage: the process is *electrogenic.* Thermodynamic analysis of the equilibrium state provides an equation to calculate the voltage for a given set of conditions; this is the equilibrium potential, $E_{rev}$ (for a $Na^+$-coupled symporter transporting a divalent anion, Equation 1, for derivation see Materials and methods):

$$E_{rev} = -\frac{60mV}{\frac{n}{m}-2}\left(\frac{n}{m}\log\frac{[Na^+]_{in}}{[Na^+]_{out}}+\log\frac{[S]_{in}}{[S]_{out}}\right)$$

where n is the number of $Na^+$ ions transported per cycle, m is the number of substrate molecules transporter per cycle, $[S]_{in}$, $[S]_{out}$, $[Na^+]_{in}$, and $[Na^+]_{out}$ are the concentrations of $Na^+$ and substrate inside and outside the vesicle, and $z_s$ is the substrate charge. When the membrane voltage differs from $E_{rev}$, substrate flux occurs; when the voltage is equal to $E_{rev}$, no flux occurs (*Figure 1*). For $Na^+$ coupled transporters, the reversal potential depends only on the substrate (S) and ion ($Na^+$) concentration gradients, the coupling stoichiometries of substrate (m) and ion (n), and the charges of both ions. Thus, measurement of the reversal potential in a known set of ion and substrate gradients uniquely determines the stoichiometric ratio, m/n. Our method takes advantage of this, measuring substrate flux at a series of voltages to find one potential where we observe no net flux; this is $E_{rev}$.

Determining $E_{rev}$ requires a series of flux measurements in the presence of constant ion gradients but with varying electrical potentials. It is critical to bracket the reversal potential with both inward and outward substrate fluxes; we thus confirm that the absence of flux reflects equilibrium rather than the absence of transport activity. In traditional flux measurements, radiolabeled substrate is introduced either inside or outside the proteoliposomes, permitting flux measurements in only one direction. Here, in contrast, we add labeled substrate at known concentrations to both sides of the membrane, permitting the system to generate all three required flux conditions, inward, outward and none.

Although this method is applicable to all electrogenic secondary transporters, we developed this approach primarily for structurally characterized proteins. By and large, these proteins have been selected for their high expression levels, ready purification, and relative stability-properties that also favor reconstitution into proteoliposomes for functional assay. Reconstituted systems yield clean flux measurements uncontaminated by the activities of native transporters or channels. Our first target was VcINDY, a $Na^+$-coupled succinate transporter whose homologs in higher organisms are important for metabolic regulation and organic acid metabolism (*Bergeron et al., 2013*). The VcINDY structure is known, demonstrating a novel protein fold (*Mancusso et al., 2012*). We recently characterized the protein's functional properties, revealing an electrogenic transport cycle in which three or more $Na^+$ ions are coupled to the transport of a doubly charged succinate ion (*Mulligan et al., 2014*). However, previous experiments could not accurately specify VcINDY's transport stoichiometry, making it an ideal test case for our new method.

In practice, we set our electrochemical gradients by loading the proteoliposomes with a constant internal ion and (radiolabeled) substrate concentration, then diluting them into external buffers containing constant (radiolabeled) substrate and $Na^+$ concentrations (*Figure 2a*). For t = 0 timepoints, we collected the loaded proteoliposomes by rapid filtration prior to exposing them to any external buffer, and measured the internalized [³H]-succinate by scintillation. For later timepoints, we diluted proteoliposomes from the same batch into external buffer containing the desired external substrate and $Na^+$. We monitored flux over time by collecting the proteoliposomes using rapid filtration and measuring the change in internalized [³H]-succinate.

Varying voltages were imposed in the presence of constant internal [$K^+$] by varying external [$K^+$] and adding valinomycin (*Mueller and Rudin, 1967*). Valinomycin is a $K^+$-selective ionophore that

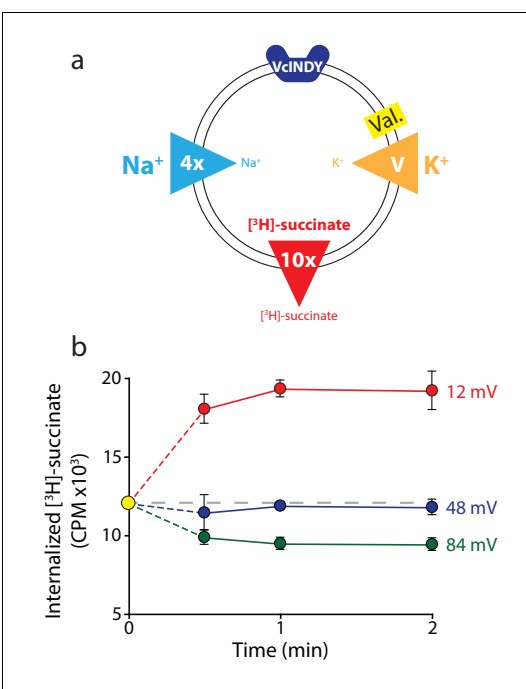

**Figure 2.** Direction of substrate flux can be controlled by the magnitude of the membrane potential. (**a**) Schematic of experimental conditions with outwardly directed [³H]-succinate gradient (red arrow) and inwardly directed $Na^+$ gradient (blue arrow) in proteoliposomes containing VcINDY. While the succinate and $Na^+$ gradients are kept constant, the $K^+$ gradient (orange arrow) was varied ('V') in the presence of valinomycin (Val.) to set the membrane potential. The direction of the arrow indicates the direction of the gradient. (**b**) Internalized [³H]-succinate (CPM) measured over time in the presence of three applied voltages; 12 mV (red), 48 mV (blue), and 84 mV (green). The yellow datapoint indicates the level of internal [³H]-succinate prior to start of the reaction. Grey dashed line indicates the initial level of internal counts. Exact buffer conditions are detailed in *Supplementary file 1*. Triplicate datasets are shown and error bars represent S.E.M. This experiment was reproduced on three separate occasions. DOI: 10.7554/eLife.21016.003

makes the membrane highly permeable to $K^+$ and brings the membrane voltage to $E_K$, the Nernst potential for potassium. We assume that the $K^+$-conductance thus induced is much higher than the conductance due to the transporters in the proteoliposomes, an assumption that has been confirmed in previous studies using valinomycin in this fashion (*Graves et al., 2008*).

Pilot experiments demonstrated the feasibility of this approach. Under a fourfold outwardly directed $Na^+$ gradient, and a tenfold inwardly directed [$^3$H]-succinate gradient, we applied three voltages with $K^+$ and valinomycin in separate reactions (*Figure 2a*). All internal and external solutions within a given experiment were osmotically balanced with choline Cl, and all solutions were buffered with 20 mM Tris/HEPES pH 7.5 (internal and external buffer compositions are detailed in *Supplementary file 1*). We observed influx of radiolabeled substrate at +12 mV, and efflux at +84 mV (*Figure 2b*). At +48 mV no net flux occurred over the full 2-min time course, indicating that 48 mV is close to the reversal potential for VcINDY in this set of concentration gradients (*Figure 2b*). These experiments demonstrate that flux direction can be dictated by voltage alone, with both influx and efflux observed at voltages bracketing the reversal potential.

We sought to definitively establish VcINDY's stoichiometry by setting up conditions where the $E_{rev}$s predicted by different candidate stoichiometries differed enough to be resolved in our system. To this end, we chose gradients such that $E_{rev}$ is predicted at 0 mV if the stoichiometry is 1:1 ($Na^+$: succinate$^{2-}$), 62 mV if the stoichiometry is 3:1, and 41 mV if the stoichiometry is 4:1 (2:1 would result in electroneutral transport, in disagreement with previously published results (*Mulligan et al., 2014*), *Figure 3a*). These voltages are sufficiently separated that we can confidently set them using our relatively crude valinomycin/$K^+$ voltage clamping system. Under these conditions, we observed influx at both 0 mV and +47 mV, the reversal potentials calculated for 1:1 and 4:1 stoichiometries respectively (*Figure 3b*). Because flux occurred at these potentials, they can be eliminated as candidate coupling stoichiometries. In contrast, we saw no net flux +62 mV, the calculated reversal potential for a 3:1 stoichiometry for VcINDY in this set of gradients (*Figure 3b*). We confirmed that efflux could occur in these liposomes, with robust exit of labeled substrate from the proteoliposomes at +80 mV. These results strongly point to a transport stoichiometry of 3 $Na^+$: 1 succinate for VcINDY.

We substantiated this stoichiometry with another set of experiments, this time with a negative predicted reversal potential (*Figure 3—figure supplement 1a and b*). Here again, we observe flux at voltages corresponding to $E_{rev}$ for 4:1 stoichiometry, but no flux at −42 mV, $E_{rev}$ for a 3:1 stoichiometric ratio, at least over the first 30 s of the experiment (*Figure 3—figure supplement 1b*). Over longer times, we consistently observed a decrease in internalized radioactivity, perhaps representing a substrate leak due to the negative potential (see Discussion), although no such leak was observed in protein-free liposomes (*Figure 3—figure supplement 1c*).

To summarize our results under all the tested conditions, we plotted the data from each of the three sets of gradients on a voltage-$\Delta$CPM (counts per minute) plot, analogous to a current-voltage relation in an electrophysiological experiment (*Figure 4*). For each set of gradients, we observe no net flux at the reversal potential predicted for a 3:1 coupling stoichiometry, while flux occurs at all potentials calculated for alternate candidate stoichiometries. The consistency of these results under widely varying conditions argues strongly that they reflect an actual equilibrium measurement, and are not skewed by leaks or other artifacts. Thus, they conclusively reveal that the coupling stoichiometry of VcINDY is 3:1 $Na^+$:Succinate$^{2-}$.

A key aspect of the previous experiments is the accurate setting of concentration gradients. While it can be assumed the freeze/thaw cycles combined with extrusion are sufficient to equilibrate the internal buffer, it is prudent to test this. We designed an experiment similar to the one described previously to determine the internal radiolabeled substrate concentration. To this end, we took advantage of the properties of Equation 1. If we set the voltage to 0 mV, then rearrange, the equation becomes (see Materials and methods for derivation):

$$\frac{[Na^+]_{in}}{[Na^+]_{out}} = \left( \frac{[S]_{out}}{[S]_{in}} \right)^{\frac{n}{m}}$$

Thus at V = 0, two conditions lead to equilibrium, and therefore no flux: either when the gradients of substrate balance that of $Na^+$ (to the power of the stoichiometric ratio), or when both substrate and $Na^+$ are both at equal concentrations inside and outside the proteoliposomes. We used the latter case to determine whether we are accurately setting the internal [S]. We performed a series of

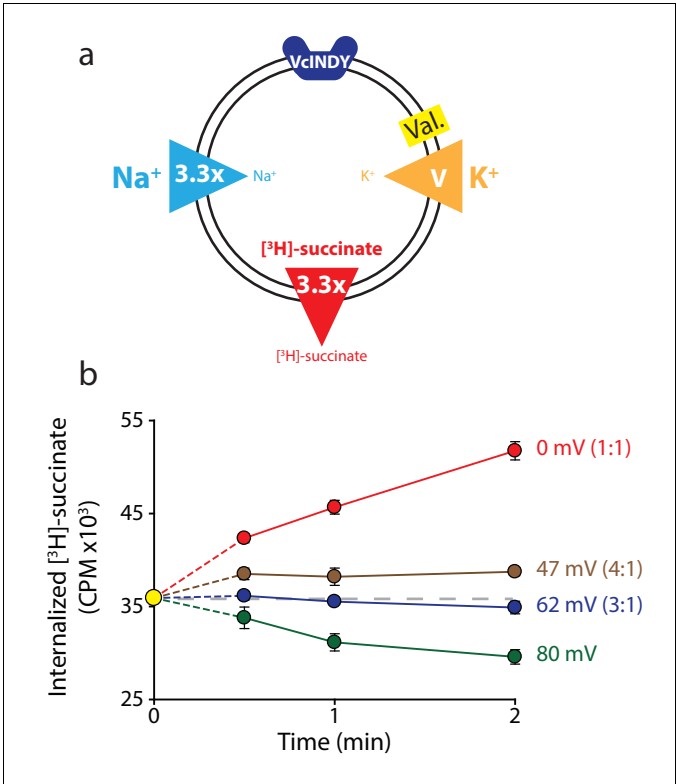

**Figure 3.** Reversal potential for VcINDY transport suggests a 3:1 coupling stoichiometry. (**a**) Schematic describing experimental conditions as in **Figure 2a**. (**b**) Internalized [³H]-succinate (CPM) over time in the presence of four different voltages: 0 mV (red), 47 mV (brown), 62 mV (blue), and 80 mV (green). The coupling stoichiometry (Na⁺: succinate²⁻) for each possible reversal potential is shown in parentheses for each applied voltage. The 80 mV condition, which is not the calculated reversal potential for any candidate coupling stoichiometry, serves as a proof of flux reversal. Grey dashed line indicates the initial level of internal counts. The exact buffer conditions used in this experiment are detailed in **Supplementary file 1**. Each experiment was performed in triplicate and error bars represent S.E.M. This experiment was performed on three occasions and found to be reproducible.

The following figure supplement is available for figure 3:

**Figure supplement 1.** Negative membrane potentials suggest a 3:1 coupling stoichiometry for VcINDY.

experiments at 0 mV, in each case attempting to set equal [Na⁺] on both sides and the internal [S] to 1 μM using freeze-thaw/extrusion; we then varied external [S] and monitored direction of flux (**Figure 4—figure supplement 1a**). In this system, zero flux will occur when internal and external substrate concentrations are equal. Indeed, we observed flux at substrate concentrations bracketing our supposed internal concentration, but no net flux when [S] = 1 μM outside the proteoliposomes, exactly the presumed internal concentration (**Figure 4—figure supplement 1b**). These results confirm that the internal buffer contains the desired concentration of S, and, we infer, the desired concentration of Na⁺, validating our procedures for setting the internal buffer concentrations.

We sought to establish the generality of our method by determining the stoichiometry of a transporter with a known coupling ratio that differs from that of VcINDY. We chose vSGLT, a Na⁺:galactose symporter from *Vibrio parahaemolyticus*, which transports Na⁺ and galactose with a ratio of 1:1 (**Turk et al., 2000**). Because the substrate (galactose) is neutral, the equation that describes the relationship between coupling stoichiometry and reversal potential is altered slightly.

$$E_{rev} = -60\left(\frac{m}{n}\log\frac{[G]_{in}}{[G]_{out}} + \log\frac{[Na^+]_{in}}{[Na^+]_{out}}\right)$$

We used essentially the same procedure as that for the VcINDY reversal potential experiments,

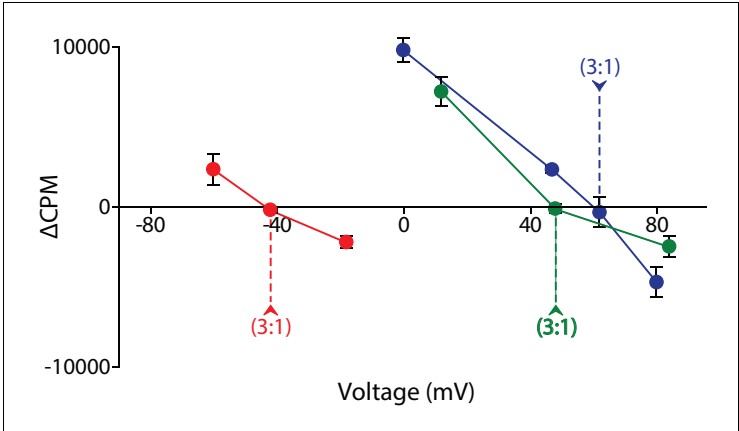

**Figure 4.** Voltage-ΔCPM plot of VcINDY transport for three different sets of substrate and coupling ion gradients. ΔCPM for three sets of gradients plotted as a function of voltage. ΔCPM values were calculated by subtracting the CPM at 1 min (green and blues traces) and 30 s (red trace) from the initial counts for each voltage tested (Green data from *Figure 2*, Blue data from *Figure 3*, Red data from *Figure 3—figure supplement 1*). The reversal potentials for each gradient set are indicated by the dashed line and each represents a 3:1 coupling stoichiometry. Triplicate data sets are shown and error bars represent S.E.M.

The following figure supplement is available for figure 4:

**Figure supplement 1.** Determining the internal concentration of succinate using flux equilibrium.

with some minor modifications due to differences in the transport properties between the proteins. vSGLT has a relatively low affinity for galactose ($K_m$158 µM) (*Turk et al., 2000*), and [3]H-labeled galactose is not available at a high enough concentration to get to measureable levels in the transport reaction. We therefore used [14C]-galactose in our assays (*Figure 5a*). Additionally, unlike for VcINDY, we observed significant variation in the starting levels of internal [14C]-galactose (t = 0) for each voltage, despite each sample originating from the same batch of proteoliposomes. To account for this variation in t = 0 values between different voltages, we normalized the dataset for each voltage by its 5 s time-point (*Figure 5b*). Lastly, due to either leakage or run-down of gradients, we observed a decrease in internalized [14C]-galactose in all tested voltages after 30 s. However, we obtained sufficient signal by 30 s to determine the coupling stoichiometry. From these experiments, we were able to confirm the reported coupling stoichiometry of 1:1 Na$^+$:galactose for vSGLT (*Figure 5b and c*). This result not only confirms previous observations for vSGLT, but also serves as a verification of the validity and practicality of this method we have devised.

## Discussion

We report here a new method for determining the coupling stoichiometry of electrogenic secondary transporters using radiolabeled substrate flux assays with purified, reconstituted protein. This method uses the thermodynamic measure of reversal potentials to calculate and test candidate coupling stoichiometries. Using this method, we report definitively that VcINDY has a 3:1 (Na$^+$:succinate$^{2-}$) coupling stoichiometry. In addition, we validated this approach by investigating the coupling ratio of vSGLT, a transporter with a known coupling stoichiometry of 1:1 (Na$^+$:galactose), which we verify here.

Our aim in developing this method was to redress the dearth of widely applicable, robust, and accurate methods for determining the coupling stoichiometry of secondary active transporters. The reversal potential-based method we describe is based on routine radiolabeled substrate flux assays; it is therefore accessible to any lab already undertaking basic transport assays. In theory, this method can be used for any secondary transporter for which radiolabeled substrates are available, although it will be most applicable to Na$^+$-coupled transporters, due to other viable options available for proton-driven systems (*Graves et al., 2008*; *Parker et al., 2014*).

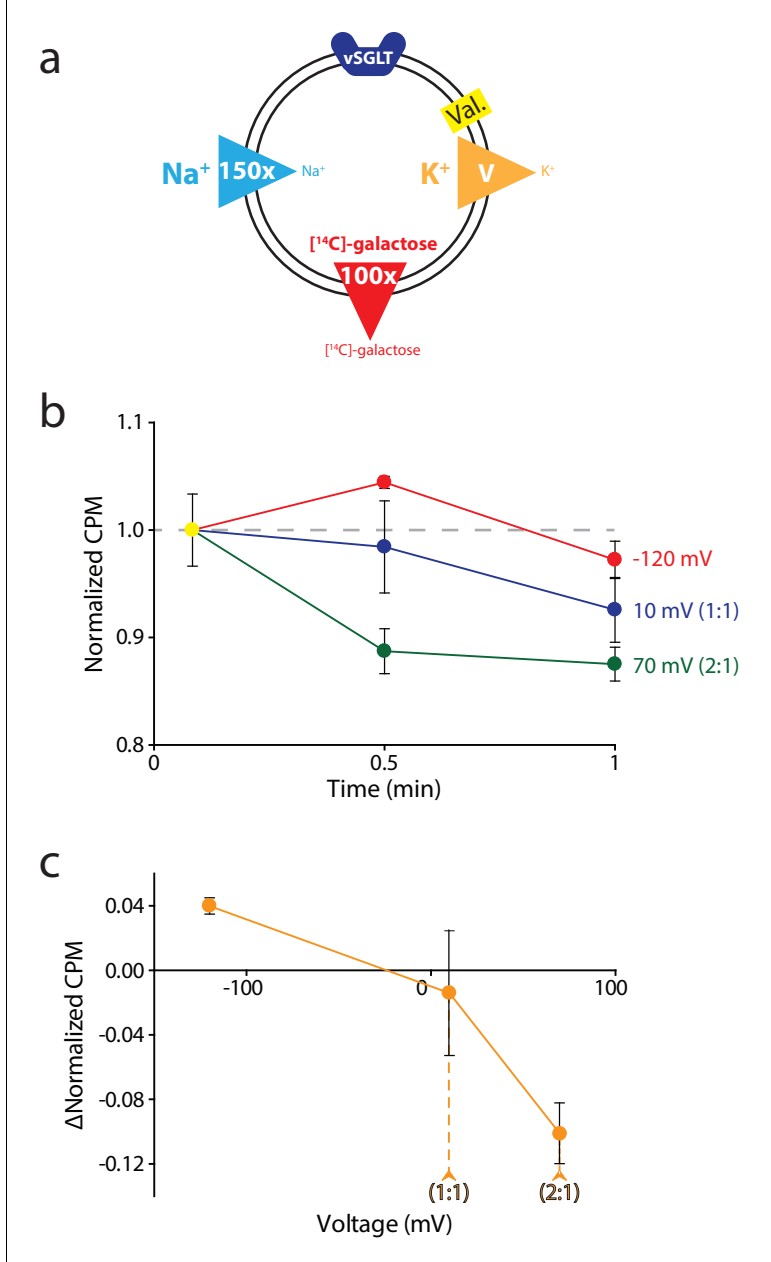

**Figure 5.** Reversal potential for vSGLT transport indicates a 1:1 coupling stoichiometry. (**a**) Schematic describing experimental conditions as in **Figure 2a.**, except with an outwardly directed [14C]-galactose gradient instead of succinate. (**b**) Internalized [14C]-galactose measured over time in the presence of three different voltages: −120 mV (red), 10 mV (blue), and 70 mV (green). The exact buffer conditions used in this experiment are detailed in **Supplementary file 1**. (**c**) Voltage-ΔCPM plot of the data in part (**b**). ΔCPM was calculated by subtracting the CPM values at 30 s from the normalized y-intercept. Numbers in parentheses represent coupling stoichiometry (Na+: galactose) for each membrane potential. Triplicate datasets are shown and the error bars represent S.E.M. This experiment was reproduced on two separate occasions.

The approach we describe offers a superior alternative to currently available methodologies; the model-dependent Hill equation approach, which can be inaccurate; and the direct measurement of 22Na+ flux, which is inconvenient, expensive, and prone to technical difficulties (**Groeneveld and Slotboom, 2010**). Another approach that has been used to determine transporter coupling ratios is the static head method (**Turner and Moran, 1982**). As in our reversal potential-based method, the

static head method sets out to find conditions in which there is no net flux of substrate by balancing oppositely directed substrate and coupling ion gradients at 0 voltage. In the static head method, accurately balancing the substrate and coupling ion gradients becomes difficult when multiple ions are coupled to substrate transport; a prime example is the situation with VcINDY where three $Na^+$ ions are coupled to the transport of one succinate molecule. In this case, to balance a 10:1 outwardly directed substrate gradient, a 1:1000 inwardly directed coupling ion gradient is required. Under these conditions, even very small leaks can become substantial problems and errors in buffer composition are magnified. A trace leak in such a case would lead to an underestimation of the concentrating power of the transporter (*Parker et al., 2014*). In contrast, the reversal potential-based method uses moderate substrate and coupling ion concentrations, thus sidestepping this issue.

To our knowledge, this is the first application of reversal potential measurements to determine transporter coupling ratios in proteoliposomes. Measuring transport phenomena in proteoliposomes has several advantages over other experimental systems available for bacterial and archaeal transporters, such as, whole cells or membrane vesicles. First, accurate stoichiometry measurements cannot be performed in whole cells, where the experimenter has little control over the internal solutions. Close control of the electrochemical gradients is a key requirement to the success of our reversal potential-based method (as well as the static head method). While a reversal potential-based experiment can feasibly be performed using native membrane vesicles, as was the case for the cardiac $Na^+/Ca^+$ exchanger (*Reeves and Hale, 1984*), a proteoliposome-centered method is more broadly applicable as it circumvents some of the limitations of using membrane vesicles. The paramount issue with using vesicles derived from native membranes is the presence of endogenous transporters and membrane associated enzymes whose activity may obscure the activity of the transporter of interest. In proteoliposomes, as the protein of interest is reconstituted in isolation, the user can be absolutely certain that only one substrate and coupling ion conducting protein is present in the membrane. Multiple routes for substrate and coupling ion movement will confound analysis and lead to inaccurate coupling ratios. Finally, if necessary, the lipid composition and transport protein density can be modulated in proteoliposomes to optimize transport and minimize substrate and coupling ion leak (*Tsai and Miller, 2013*). Such modifications are not possible with membrane vesicles where the user is saddled with the protein and lipid that is present in the source material.

While our reversal potential-based method has advantages over other available methods, it is not without its limitations. Most obviously, this method is only applicable to electrogenic transporters. Clearly, the method is most useful for $Na^+$-coupled transporters; if the transporter also couples to $K^+$, for example, the use of valinomycin/$K^+$ to set voltage will be compromised. In theory, pH gradients combined with the proton ionophore carbonyl cyanide m-chlorophenyl hydrazone (CCCP) could be used to set the potential, but we have not attempted this. However, among the known bacterial transporters (for which this method is intended) few are known to utilize $K^+$. For those that do, the $K^+$ concentrations required for transporter effects are quite high and could be avoided in practice (*Rübenhagen et al., 2001*; *Billesbølle et al., 2016*). Also, some transporters manifest an uncoupled $Cl^-$ conductance (*Ryan and Mindell, 2007*). Depending on the relative fluxes through this pathway compared with the coupled transport pathway this could also perturb stoichiometry measurements. Again, we believe that these complications are rare, and the method will be useful for the vast majority of $Na^+$-coupled prokaryotic transporters.

Substrate leaking from the proteoliposomes could also compromise the measurement of reversal potentials; however, this is not insurmountable. Indeed, when we tested exclusively negative voltages with VcINDY (*Figure 3—figure supplement 1*), we observed a systemic drop in internalized radiolabeled substrate after the 30 s timepoint that we speculatively attribute to substrate leak that develops over time. The leak is reproducible under these conditions, but not seen in other experiments (not shown) at negative voltages. Control experiments with protein-free liposomes show no evidence of succinate loss (*Figure 3—figure supplement 1c*), suggesting that the effect is mediated by the VcINDY protein. Despite this small leak, we were still able to unambiguously determine a 3:1 coupling ratio for VcINDY by using the 30 s timepoints only, before the leak has substantially contributed to the observed efflux, and combining these measurements with those performed under other conditions (*Figure 4*).

A key assumption we make in our assay is that we can accurately set the membrane potential with $K^+$/valinomycin and that the voltage is maintained throughout the entirety of our assay. Although we have not independently verified that we are able to achieve the voltages we presume

to be setting, the 'chemical patch clamp' method has been used extensively (*Graves et al., 2008*; *Parker et al., 2014*). Additionally, it is very unlikely that we would have observed consistent reversal potential results over a large span of voltages if we were not accurately clamping the membrane potential. Note that it is also important to consider the limited internal volume of the vesicles used in these experiments. For a 400 nm vesicle containing 1 µM substrate, that represents around 20 molecules of substrate—at lower substrate concentrations (or in smaller vesicles) there could easily be only one or a few molecules per vesicle. Averaged over a large population (billions used in these experiments), this may still be sufficient to observe substrate efflux at appropriate vesicles, but problems or inconsistencies could arise from such low substrate numbers.

Unlike other methods for determining coupling ratios, the reversal potential-based method does not theoretically require any prior knowledge of the transport mechanism. However, knowing certain mechanistic details will make determining the coupling stoichiometry more practical. As a starting point, it is essential to know what ions are coupled to substrate transport, since all these will contribute to determining the reversal potential. For example, if a proton is surreptitiously coupled to transport then failing to account for it in the $E_{rev}$ calculations would clearly lead to incorrect results. Knowing the charge of the substrate and the number of substrates that are transported per cycle is also beneficial, as it will decrease the number of assays required to obtain the coupling ratio. In our experience, the transporter in question will ideally have a reasonably high apparent affinity for the radiolabeled substrate (low µM range). As we observed for vSGLT (see *Figure 5*), large substrate gradients may be necessary to achieve resolvable voltage differences and, as a consequence of this, the substrate concentration on one side of the membrane will, by necessity, be quite low. If a low-affinity transporter is being probed, then the substrate concentration on one side of the membrane may be too far below the $K_m$ to observe flux in this direction in the timeframe of the experiment. As a starting point, we recommend that the substrate concentrations on either side of the membrane should be kept to within 1/10th of the $K_m$ to achieve measureable transport rates within the duration of the assay. It is also important to ensure that there is sufficient substrate in the vesicles. At 1 µM, there are on the order of 10–20 substrate molecules in each vesicle. Our efflux results demonstrate that this is sufficient to make the needed measurement (especially averaged over the $\sim 10^{10}$ vesicles in each experiment), but lower concentrations could be problematic.

VcINDY is an excellent test case for our reversal potential-based approach as the crystal structure revealed a single substrate and a single $Na^+$ ion bound (*Mancusso et al., 2012*), incongruent with the functional characterization of VcINDY, which demonstrates that at least three $Na^+$ ions are coupled to the transport of a single succinate$^{2-}$ ion (*Mulligan et al., 2014*). VcINDY, which was recently demonstrated to utilize an elevator-like protein movement to transport substrate across the membrane (*Mulligan et al., 2016*), is the only structurally characterized member of the DASS family and currently acts as an experimentally tractable model protein with which to probe the mechanism of this family, which includes several human proteins that could be targeted in the treatment of metabolic diseases and diabetes (*Bergeron et al., 2013*). It is therefore vital to thoroughly understand VcINDY's structural and energetic mechanism. We have unambiguously determined that VcINDY couples three $Na^+$ ions to the transport of a single succinate$^{2-}$, the same coupling ratio demonstrated for hNaDC1 and 3 (*Chen et al., 1998*; *Kekuda et al., 1999*), reinforcing that VcINDY is an excellent mechanistic representative of this large family. In this case, at least, the stoichiometry measured here is consistent with the value we previously estimated using measurements of Hill coefficient (*Mulligan et al., 2014*), suggesting that in this case the latter was an accurate predictor. Together with further ($Na^+$) Hill coefficient measurements at different substrate concentrations these results could be useful in probing the binding order of substrate and $Na^+$ ions (*Lolkema and Slotboom, 2015*).

The number and location of coupling ion binding sites is an essential piece of the puzzle in fully illuminating transporter mechanism. Coupling ions are often not visible in crystal structures, either due to the low resolution of most membrane protein structures or the crystal structure capturing a state in which one or more coupling ions had already been released. Computational approaches, such as molecular dynamic simulations, are powerful methods of predicting and testing, in silico, coupling ion locations. However, these procedures need experimental validation and are only as accurate as the data that is input. The reversal potential-based method we describe provides both unambiguous coupling ion stoichiometries to improve the accuracy of these computational approaches and a means of testing subsequent computational predictions. In particular, this method

will be useful in studies aiming to alter or eliminate transporter ion-binding sites, where current methods do not accurately report changes in stoichiometry.

The mechanisms of secondary transporters cannot be addressed by structural biology alone. With this in mind, we have developed a broadly applicable method to determine coupling ratios of secondary transporters. This method will allow researchers to probe the blind spots of structural methods and enhance the accuracy of ever-improving computational approaches.

## Materials and methods

Derivation of the reversal potential equation for a VcINDY, a $Na^+$-coupled succinate transporter, where succinate is transported in the $-2$ charge state.

We assume that transport occurs with a fixed stoichiometry as reflected below:

$$nNa^+_{out} + mS^{2-}_{out} \rightleftharpoons nNa^+_{in} + mS^{2-}_{in}$$

where $S$ indicates the doubly charged succinate, the subscripts *out* and *in* refer to sodium or succinate outside or inside the liposomes, n is the number of sodium ions transported per cycle and m is the number of succinate ions. We seek to determine n/m the stoichiometric ratio of sodium: succinate.

For this reaction:

$$\mu_{Naout} = \mu^\circ_{Na} + RT\ln[Na^+]_{out} + z_{Na}F\Psi_{out}$$
$$\mu_{Nain} = \mu^\circ_{Na} + RT\ln[Na^+]_{in} + z_{Na}F\Psi_{in}$$
$$\Delta\mu_{Na} = \mu_{Nain} - \mu_{Naout}$$

$$\Delta\mu_{Na} = RT\ln\frac{[Na^+]_{in}}{[Na^+]_{out}} + z_{Na}F\Delta\Psi, z_{Na} = +1$$
*Similarly*,

$$\Delta\mu_S = RT\ln\frac{[S]_{in}}{[S]_{out}} + z_S F\Delta\Psi, z_S = -2$$

where μ is the chemical potential of the species, μ° is the standard-state chemical potential of the species, R is the universal gas constant, T is the temperature (in °K), F is the Faraday constant, $Z_{Na}$, and $z_S$ are the sodium and substrates charges, respectively, $\Delta\Psi$ is the voltage difference across the membrane, where $\Delta\Psi = \Psi_{in} - \Psi_{out}$. This is equivalent to the sign convention that the outside of the liposome is defined as ground ($\Psi_{out} = 0$).

At equilibrium,

$$\sum n_i\mu_i = 0$$
$$n\mu_{Nain} + m\mu_{Succin} - n\mu_{Naout} - m\mu_{Succout} = n\Delta\mu_{Na} + m\Delta\mu_{succ} = 0$$

So,

$$0 = n\left(RT\ln\frac{[Na^+]_{in}}{[Na^+]_{out}} + z_{Na}F\Delta\Psi\right) + m\left(RT\ln\frac{[S]_{in}}{[S]_{out}} + z_S F\Delta\Psi\right)$$

Rearranging and setting $Z_{Na} = 1$ and $Z_S = -2$:

$$-(n - 2m)F\Delta\Psi = nRT\ln\frac{[Na^+]_{in}}{[Na^+]_{out}} + mRT\ln\frac{[S]_{in}}{[S]_{out}}$$

Therefore, at equilibrium (with conversions to base 10 log, and approximating RT/F as 60 mV:

$$E_{rev} = \Delta\Psi = \frac{-60mV}{n-2m}\left(n\log\frac{[Na^+]_{in}}{[Na^+]_{out}} + m\log\frac{[S]_{in}}{[S]_{out}}\right)$$
$$E_{rev} = \Delta\Psi = \frac{-60mV}{\frac{n}{m}-2}\left(\frac{n}{m}\log\frac{[Na^+]_{in}}{[Na^+]_{out}} + \log\frac{[S]_{in}}{[S]_{out}}\right)$$

This is the desired result, yielding the equilibrium, or reversal, potential, $E_{rev}$, in terms of the values of the $Na^+$ and succinate gradients and the stoichiometric ratio n/m. Note that if n/m = 2 the transporter would be electroneutral and the equation becomes undefined.

In the case that the voltage is zero,

$$0 = \frac{-60mV}{\frac{n}{m}-2}\left(\frac{n}{m}\log\frac{[Na^+]_{in}}{[Na^+]_{out}} + \log\frac{[S]_{in}}{[S]_{out}}\right)$$

Thus,

$$-\frac{n}{m}\log\frac{[Na^+]_{in}}{[Na^+]_{out}} = \log\frac{[S]_{in}}{[S]_{out}}$$

or

$$\frac{[Na^+]_{in}}{[Na^+]_{out}} = \left(\frac{[S]_{out}}{[S]_{in}}\right)^{\frac{n}{m}}$$

## Protein expression and purification

VcINDY was expressed and purified using the protocol developed by Mancusso et al (*Mancusso et al., 2012*). Briefly, BL21-AI (Life Technologies Carlsbad, CA) was transformed with a pET vector containing the VcINDY gene in-frame with an N-terminal deca-histidine tag. Cells were cultured in LB supplemented with 300 µg/ml kanamycin to until $A_{600}$ 0.8 was reached and then cooled in an ice-water bath for 20 min. Protein expression was induced by adding 0.1 M IPTG and 6.6 mM L-arabinose to the cultures. Cultures were incubated overnight at 19°C, then harvested and lysed using a homogenizer (EmulsiFlex-C3; Avestin). The lysate was clarified and membrane vesicles were isolated by ultracentrifugation. Membrane vesicles were then resuspended in Purification Buffer (PB) containing 50 mM Tris-HCl, pH 8, 100 mM NaCl and 5% (vol/vol) glycerol and solubilized by adding n-dodecyl-$\beta$-D-maltoside (DDM; Anatrace) to a final concentration of 20 mM. Insoluble matter was removed by ultracentrifugation, and the soluble supernatant was incubated with TALON-affinity resin (Takara Bio Inc.) overnight at 4°C. Weakly bound impurities were eluted with two consecutive 20 column volume washing steps, the first with PB supplemented with 2 mM DDM and 10 mM imidazole and the secondwith buffer containing 20 mM imidazole. VcINDY was eluted, and the affinity tag removed, by incubating the protein/resin mixture with 10 µg/ml trypsin for 1 hr at 4°C.

vSGLT was expressed and purified as detailed previously (*Faham et al., 2008*). TOP10 (Life Technologies Carlsbad, CA) cells, transformed with a pBAD vector containing the vSGLT gene in-frame with a C-terminal Histidine tag and the mutation A423C (*Xie et al., 2000*), were grown at 37°C in TB supplemented with 100 µg/ml ampicillin to an $A_{600}$ of 1.8. Expression was induced by adding 0.66 mM L-arabinose. Cultures were then incubated at 29°C for a further 4 hr. Membrane vesicles were prepared as for VcINDY, then resuspended in vSGLT Purification Buffer (sgltPB) containing 70 mM Tris-HCl, pH 8, 150 mM NaCl, 20 mM imidazole, 4 mM $Na_3$Citrate, 5 mM $\beta$-mercaptoethanol, and 6% (vol/vol) glycerol. The membrane vesicles were solubilized by adding DDM to a final concentration of 40 mM. Solubilized protein was separated from the insoluble matter by centrifugation and then incubated with Ni-NTA Superflow resin (Qiagen, Germany) overnight at 4°C. The resin/protein mixture was washed with 20 CV of sgltPB supplemented with 3.6 mM DM and eluted with the same buffer supplemented with 180 mM imidazole.

For both VcINDY and vSGLT, affinity purified protein was concentrated and further purified using a Superdex 200 10/300 GL (GE Healthcare) size exclusion chromatography (SEC) column.

## Protein reconstitution

Purified VcINDY and vSGLT were reconstituted using a rapid dilution method (*Mulligan et al., 2009*). Briefly, 25 µg of VcINDY or 200 µg of vSGLT was diluted to 2 ml in buffer containing 20 mM Tris/HEPES, pH 7.5, 1 mM NaCl, 199 mM KCl and either 2.1 mM or 3.6 mM DM for VcINDY and vSGLT, respectively. Protein was mixed with 8 mg lipid consisting of a 3:1 mixture of *E. coli* polar lipids and POPC (Avanti Polar Lipids, Inc.). The protein/lipid mixture was incubated on ice for 10 min, then rapidly diluted into buffer containing 20 mM tris/HEPES pH 1 mM NaCl, 199 mM KCl. Proteoliposomes (PLs) were collected by ultracentrifugation, resuspended in desired internal buffer to a final concentration of 8 mg/ml lipid and freeze-thawed three times. Proteoliposomes were either stored at −80°C or used immediately. Before the proteoliposomes could be used in transport assays, they were concentrated to a final concentration of 80 mg/ml lipid using ultracentrifugation.

## Preparation of proteoliposomes for transport assays

The lumen of the proteoliposomes were loaded with the desired internal solution by firstly diluting 100 µl 80 mg/ml PLs into the desired internal buffer containing 20 mM Tris/HEPES, pH 7.5, variable

NaCl/KCl/ChCl concentrations depending on the experiment (see Results), and the desired concentration of radiolabeled substrate; for VcINDY, this was 1 µM [$^3$H]-succinate (60 Ci/mmol, 1 mCi/ml, American Radiolabeled Chemicals); and for vSGLT, this was 362 µM [$^{14}$C]-galactose (55 mCi/mmol, 0.1 mCi/ml, American Radiolabeled Chemicals). The final PL concentrations were 8.42 mg/ml and 32 mg/ml lipid for VcINDY and vSGLT, respectively. The diluted PLs were freeze-thawed three times and extruded through a 400-nm polycarbonate membrane (Whatman) to equilibrate the internal and external solutions.

## Transport assays

To start the transport assay, loaded proteoliposomes (674 µg lipid) were diluted into Reaction Buffer containing 20 mM Tris/HEPES, pH 7.5, 1 µM valinomycin, and NaCl/KCl/ChCl concentrations varied depending on the experiment (see *Supplementary file 1* for details). No additional radiolabeled substrate was added to the Reaction Buffer, so diluting the isotope-loaded proteoliposomes (which will have significant extra-liposomal radiolabeled substrate present) into the Reaction Buffer was sufficient to set the desired substrate gradient. The extent of proteoliposome dilution was therefore dictated by the succinate gradient we wished to achieve. Samples were taken at the specified time-points, rapidly filtered on 0.22-µm nitrocellulose membranes (Merck Millipore) to collect the proteoliposomes, then washed by addition of 4 ml of ice-cold Quench Buffer (20 mM Tris/HEPES, ChCl osmotically balanced to inside buffer). The filters were dried, dissolved in liquid scintillation cocktail (FilterCount, PerkinElmer), and counted on a Trilux $\beta$ counter (PerkinElmer). The initial point (t = 0) value was determined by diluting preloaded proteoliposomes directly into ice-cold 2 ml Quench Buffer, rapidly filtering and washing with 4 ml of Quench Buffer. No initial point was taken for vSGLT; instead a 5 s time-point was taken to act as the initial value.

## Acknowledgements

This work was supported by the Division of Intramural Research of the NIH, National Institute of Neurological Disorders and Stroke. We thank Jeff Abramson for the gift of vSGLT expression plasmid and Merritt Maduke, Kenton Swartz, and Michael Grabe for insightful and helpful comments on the manuscript

## Additional information

### Funding

| Funder | Grant reference number | Author |
|---|---|---|
| National Institute of Neurological Disorders and Stroke | Intramural Program | Joseph A Mindell |

The funders had no role in study design, data collection and interpretation, or the decision to submit the work for publication.

### Author contributions

GAF, Formal analysis, Investigation, Methodology, Writing—original draft, Writing—review and editing; CM, Investigation, Writing—original draft, Writing—review and editing; JAM, Conceptualization, Formal analysis, Supervision, Funding acquisition, Project administration, Writing—review and editing

### Author ORCIDs

Joseph A Mindell, http://orcid.org/0000-0002-6952-8247

## Additional files

### Supplementary files

• Supplementary file 1. Exact buffer conditions for all experiments presented. Constituent concentrations of internal and external buffers used in each experiment. ICD stands for internal concentration

determination. Columns are color-coded to match the experimental schematics presented in each figure.

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
