## [Decision Letter]

Thank you for submitting your article "Counting the cost of transport: a general method for determining secondary active transporter substrate stoichiometry" for consideration by *eLife*. Your article has been favorably evaluated by Richard Aldrich (Senior Editor) and two reviewers, one of whom is a member of our Board of Reviewing Editors. The reviewers have opted to remain anonymous.

The reviewers have discussed the reviews with one another and the Reviewing Editor has drafted this decision to help you prepare a revised submission.

The manuscript by the Mindell lab presents a method to determine the sodium ion: substrate stoichiometry in sodium coupled secondary transporters based on reversal potential measurements. The methodology is delightfully classic, and is a worthwhile addition in the repertoire of assays to study the function of bacterial transporters. The reviewers were enthusiastic, but had three key concerns:

1) The limitations of the technique need to be described more explicitly. These should include limitation only to Na-coupled transporters. Potassium coupling will be much more difficult to address because of the use of valinomycin, and chloride coupling is also potentially difficult because of leakage. The authors should emphasize that the method has been tested only for sodium coupling instead of stating that it is applicable to all electrogenic transporters. The authors should briefly discuss how uncoupled chloride conductance (observed in some transporters such as GltPh) would affect the measurements. Some transporters (e.g. MelB) can use either sodium ions or protons as coupling ion. Because the authors present the method as "generic" they should briefly discuss what are the consequences of such promiscuity. In some cases, such as in LeuT, for example, transport is also coupled to counter-transport of protons. Such coupling can lead to artificially lower coupling stoichiometry for a given transporter. What test one would run to ensure that there is no "hidden" coupling to protons?

2) The reviewers were concerned about the leakage observed at negative potential. There was a consensus that background leak should be determined, perhaps using protein-free liposomes. The authors make a strong statement about the advantages of using proteoliposomes being "uncontaminated by leak pathways" (Results, fourth paragraph), yet they clearly suffer from leaks. Furthermore, it is not clear that timepoint taken at 30s makes it possible to avoid problems caused by the leaks. Clearly the leak will be there from timepoint 0 onward. Therefore, the dashed line in Figure 3—figure supplement 1 is really very misleading. It suggests that the authors believe that there is no leakage in the first 30 s of the measurement, after which it magically starts. The interpretation of this data should be phrased much more carefully.

3) The reviewers were concerned regarding the usage of low internal substrate concentration. At 1 μm internal substrate and small liposomes (which might be on average around 200 nm) there are only a few molecules of substrate inside the liposomes and a proportion of liposomes might be empty. Unlike classical cell-based experiments, in which there are essentially "infinite" amounts of substrates and ions and steady state fluxes are measured, in the liposome experiments, stochasticity may play a significant role. It is not entirely clear whether it will affect determination of "reversal potential" (if this term is truly applicable here). The authors should discuss this point carefully. Furthermore, reviewers generally felt that it would have been worthwhile to confirm that identical results are obtained when higher internal concentrations of the substrate are used.

---

## [Author Response]

*The manuscript by the Mindell lab presents a method to determine the sodium ion: substrate stoichiometry in sodium coupled secondary transporters based on reversal potential measurements. The methodology is delightfully classic, and is a worthwhile addition in the repertoire of assays to study the function of bacterial transporters. The reviewers were enthusiastic, but had three key concerns:*

*1) The limitations of the technique need to be described more explicitly. These should include limitation only to Na-coupled transporters. Potassium coupling will be much more difficult to address because of the use of valinomycin, and chloride coupling is also potentially difficult because of leakage. The authors should emphasize that the method has been tested only for sodium coupling instead of stating that it is applicable to all electrogenic transporters. The authors should briefly discuss how uncoupled chloride conductance (observed in some transporters such as GltPh) would affect the measurements. Some transporters (e.g. MelB) can use either sodium ions or protons as coupling ion. Because the authors present the method as "generic" they should briefly discuss what are the consequences of such promiscuity. In some cases, such as in LeuT, for example, transport is also coupled to counter-transport of protons. Such coupling can lead to artificially lower coupling stoichiometry for a given transporter. What test one would run to ensure that there is no "hidden" coupling to protons?*

We agree that these points are important to address. We have added to the Discussion to consider them.

*2) The reviewers were concerned about the leakage observed at negative potential. There was a consensus that background leak should be determined, perhaps using protein-free liposomes. The authors make a strong statement about the advantages of using proteoliposomes being "uncontaminated by leak pathways" (Results, fourth paragraph), yet they clearly suffer from leaks. Furthermore, it is not clear that timepoint taken at 30s makes it possible to avoid problems caused by the leaks. Clearly the leak will be there from timepoint 0 onward. Therefore, the dashed line in Figure 3—figure supplement 1 is really very misleading. It suggests that the authors believe that there is no leakage in the first 30 s of the measurement, after which it magically starts. The interpretation of this data should be phrased much more carefully.*

The reviewers have raised an excellent point. We have edited the text to clarify the issue, and have done further experiments. We see no leak of substrate in protein-free liposomes, suggesting that the vcINDY protein itself is mediating the observed efflux. In addition, a quick and dirty experiment with different conditions also predicted to give rise to a negative reversal potential gave results again consistent with 3/1 stoichiometry and no evidence of leak, further supporting our conclusions. We are left speculating that the leak must develop over time, or that the voltage is somehow shifting during the experiment. We are still convinced of the result, given the consistency with two+ other experimental conditions, and will follow up this observation with more experiments in the future (Discussion, sixth paragraph, Figure 3—figure supplement 1).

*3) The reviewers were concerned regarding the usage of low internal substrate concentration. At 1 μm internal substrate and small liposomes (which might be on average around 200 nm) there are only a few molecules of substrate inside the liposomes and a proportion of liposomes might be empty. Unlike classical cell-based experiments, in which there are essentially "infinite" amounts of substrates and ions and steady state fluxes are measured, in the liposome experiments, stochasticity may play a significant role. It is not entirely clear whether it will affect determination of "reversal potential" (if this term is truly applicable here). The authors should discuss this point carefully. Furthermore, reviewers generally felt that it would have been worthwhile to confirm that identical results are obtained when higher internal concentrations of the substrate are used.*

The reviewers have raised an important point here, one that we are having to deal with explicitly in some other projects in the lab. However, we do not consider these issues to be critical for these experiments. A rough calculation suggests that there are about 20 molecules of substrate in each liposome on average. If these are Poisson distributed, the standard deviation is therefore about 4 molecules, so the overwhelming majority of the vesicles will have 16-24 molecules of substrate. From our data, this is clearly sufficient to provide enough internal counts that we can observe efflux at appropriate voltages. We are also aided the by the macroscopic nature of the experiment, with on the order of 10^10^ liposomes per measurement. We added a sentence to the Discussion about this (eighth paragraph).